

# ENSO statistics, teleconnections, and atmosphere-ocean coupling in the Taiwan Earth System Model version 1

Yi-Chi Wang[1], Wan-Ling Tseng[2], Yu-Luen Chen[1], Shi-Yu Lee[1*], Huang-Hsiung Hsu[1], Hsin-Chien Liang[1]

1Research Center for Environmental Changes, Academia Sinica, Taipei, Taiwan

2International Degree Program in Climate Change and Sustainable Development, National Taiwan University, Taipei, Taiwan

*Correspondence to*: Shi-Yu Lee (shihyu@gate.sinica.edu.tw)

**Abstract.** This study provides an overview of the fundamental statistics and features of the El Niño Southern Oscillation (ENSO) in the historical simulations of the Taiwan Earth System Model version 1 (TaiESM1). Compared with observations,
TaiESM1 can reproduce the fundamental features of observed ENSO signals, including seasonal phasing, thermocline coupling with winds, and atmospheric teleconnection during El Niño events. However, its ENSO response is approximately two times stronger than the observance in the spectrum, resulting in powerful teleconnection signals. The composite of El Niño events shows a strong westerly anomaly extending fast to the east Pacific in the initial stage in March, April, and May, initiating a warm sea surface temperature anomaly (SSTA) there. This warm SSTA maintains through September, October, and November
(SON) and gradually diminishes after peaking in December. Analysis of wind stress-SST and heat flux-SST coupling proposes that biased positive SST-shortwave feedback contributes significantly to the strong warm anomaly over the eastern Pacific, especially in SON. Our analysis demonstrates TaiESM1's capability of simulating ENSO—a significant tropical climate variation on interannual scales with strong global impacts, and provides insights into mechanisms in TaiESM1 related to ENSO biases, laying the foundation for future model development to reduce uncertainties in TaiESM1 and climate models in general.

**1 Introduction**

The El Niño Southern Oscillation (ENSO) is the primary mode of interannual and decadal climate variability in the tropics (Glantz, 2001; McPhaden et al., 2006). It also affects climate variations in subtropical and mid-latitude regions across both hemispheres, as we, through teleconnection of Rossby waves (Diaz et al. 2001; Yeh et al. 2018). Therefore, its prediction is an essential part of global climate prediction on these scales (Latif et al., 1998). Additionally, ENSO is a crucial metric for
climate model evaluation, especially for atmosphere–ocean coupling and associated physical feedbacks (Planton et al., 2021). Many studies have reported that coupled model intercomparison project (CMIP) models have successfully represented the basic features of observed ENSO, such as recognizable ENSO lifecycle and SST pattern over the tropical central and eastern Pacific (Guilyardi et al., 2009, 2020; Lloyd et al., 2009, 2011; Bellenger et al., 2014). However, many model biases are also



found in CMIP6 models (Capotondi et al. 2020; Chen and Jin 2021; Beobide-Arsuaga et al. 2021), resulting in 30%–50%

uncertainties in future ENSO projection (Beobide-Arsuaga et al. 2021).

The paradigm based on theory and observations depicts ENSO as a closely coupled oscillation system between the ocean and the atmosphere (Philander, 1989; Jin, 1997; Latif et al., 1998; McPhaden et al., 1998, 2006; Neelin et al., 1998; Wang and Picaut, 2013; Wang, 2018). An El Niño event begins from a westerly wind initiated over the west equatorial Pacific, typically in the spring of the first year (i.e., March, April, and May in year 0; $MAM^0$). The westerly wind drives more warm

water towards east and gradually warms the central Pacific. Around summer, eastward propagating oceanic Kelvin waves are triggered over the central Pacific, reducing the upwelling at the tropical east Pacific, deepening the thermocline, and warming the sea surface temperature (SST). Consequently, the zonal SST gradient and the easterly wind in the tropical Pacific are reduced. Such a retreat of the easterly wind further reduces the SST gradient, causing the so-called Bjerknes feedback between the easterly wind and the SST gradient (Bjerknes, 1969; Cane, 2005). Through this feedback, the warm SST increases and

reaches a maximum around the following winter (i.e. December of the year 0 and January and February of the year 1; $DJF^{+1}$). Furthermore, following the warm SST anomaly (SSTA), the center of deep convection activity shifts toward the central Pacific, increasing latent heat flux and reducing shortwave heat flux into the ocean surface through deep cloud cover. Such seasonal phase-locking of an El Niño event is a crucial characteristic of the observed ENSO.

In contrast to the coupling nature of the atmosphere and ocean found in ENSO observations, the atmospheric feedback

are found to dominate the modeled ENSO frequency and amplitude in CMIP models (Guilyardi et al., 2009; Lloyd et al., 2009, 2011; Bellenger et al., 2014; Beobide-Arsuaga et al., 2021). They noticed that the CMIP models tend to simulate a weaker Bjerknes feedback, namely, a weaker SST warming and westerly wind coupling. Furthermore, the heat flux-SST feedbacks are overemphasized in simulated ENSO dynamics, especially for the shortwave heat flux-SST feedback. Such overemphasized heat flux-SST feedback compensates for the weaker warming from the Bjerknes feedback, producing a seemingly realistic

ENSO warming in CMIP models (Bayr et al., 2019). The biased Bjerknes and heat flux feedbacks are later found to be related to the biases of the seasonal variations of Walker circulation (Bayr et al., 2019, 2018). Such complexity resulting from intertwined atmospheric–ocean feedbacks makes it challenging for model developers to improve ENSO simulations without fully understanding how these mechanisms are represented in the coupled models.

Taiwan Earth System Model version 1 (TaiESM1; Lee et al. 2020) is the earth system model developed at the Research

Center for Environmental Changes (RCEC), Academia Sinica. It participates in CMIP6 intercomparison activity and has been used in studying major climate variabilities and regional climate features (e.g. Park et al. 2020; Chen and Jin 2021). While its overall performance of climate mean states and major variations has been evaluated and documented in Wang et al. (2021), in this study, we conducted more comprehensive investigation in ENSO's fundamental features and statistics in historical TaiESM1 simulations. We noticed that the ENSO amplitude increased significantly compared with the observations, with

intense and prolonged warm SST from May to December. Especially over the eastern Pacific, the early onset of warming and sustained warming in September, October, and November ($SON^0$) before peaks in December are the two primary prominent biases. We further analyzed the physical processes associated with ENSO's strong warming biases and found it is due to the



biased positive feedback between SST-shortwave surface fluxes over the eastern equatorial Pacific. Such feedback is primarily attributed to the prevailing low clouds overlying the cold tongue region with large cold biases in SON[0]. Our results provide

the baseline of ENSO performance of TaiESM1, and suggest the current ENSO biases in TaiESM1 is intertwined with biases of mean state and seasonal variation of the tropical climate system. The remainder of this study includes the following. Section 2 describes the TaiESM1 and observational dataset used for model evaluation and the methodology for analyzing ENSO. Section 3 documents the basic characteristics of ENSO simulated in TaiESM1. More analysis focuses on seasonal variation of El Nino events in TaiESM1 and associated biases in Section 4. The study is concluded with a summary and discussion in

Section 5.

## 2 Data, Models, and Methodology

Based on the Community Earth System Model version 1.2.2 (CESM1.2.2; Hurrell et al. 2013) developed by the National Center for Atmospheric Research and sponsored by the National Science Foundation and the Department of Energy in the United States, TaiESM1 includes several physical schemes developed in-house in RCEC. These designs include convective

triggering (Wang et al., 2015b), radiation parameterization of 3-dimension topography (Lee et al., 2013), an aerosol scheme (Chen et al., 2013), and a probability density function-based cloud fraction scheme (Shiu et al., 2020). The ocean component is the same as CESM1 using the Parallel Ocean Program version 2 (POP2; Smith et al. 2010). An overall evaluation of climate variability in TaiESM1 shows that the simulated ENSO features stronger SST warming and atmospheric teleconnection compared with the base model CESM1 (Wang et al., 2021). In this study, we have conducted a more in-depth analysis of

ENSO's fundamental features and statistics and identified the physical processes of ENSO biases within TaiESM1. The historical simulation of TaiESM1 from 1850 to 2014, driven by the forcing designed by the CMIP6, is analyzed. The historical run is initiated from the pre-industrial control run of TaiESM1, with a horizontal resolution of 0.9° latitude × 1.25° longitude and 30 vertical layers.

We evaluated the model's performance using the atmospheric variables from the Collaborative Reanalysis Technical

Environment Multireanalysis Ensemble version 2 (MRE2; Potter et al. 2018). The MRE2 is a product of the ensemble average of seven reanalysis products, including CFSR (Saha et al., 2010), ERA-Interim (Dee et al., 2011), MERRA (Rienecker et al., 2011), MERRA-2 (Gelaro et al., 2017), JRA-25 (Onogi et al., 2007), JRA-55 (Kobayashi et al., 2015), and 20CRv2c (Compo et al., 2011). Studies have found that the ensemble average can reduce the errors of individual reanalysis for selected atmospheric variables (Potter et al. 2018). For those variables not provided in MRE2, such as cloud cover, we used the ECMWF

reanalysis version 5 (ERA5), the most up-to-date reanalysis produced by ECMWF (Hersbach et al., 2020). While previous studies have identified differences in air-sea feedbacks among reanalysis datasets, the ensemble mean of multiple reanalysis datasets can be used as the best estimate by reducing random errors through averaging (Kumar and Hu, 2012). We also obtained precipitation data from the Global Precipitation Climatology Project (GPCP V2; Adler et al., 2003, Huffman et al., 2009), SST data from the Extended Reconstructed SST version 5 (ERSSTv5, Huang et al. 2017), and subsurface ocean data, such as sea



surface height (SSH) and potential subsurface temperature, from the Simple Ocean Data Assimilation version 3.3.2 (SODA v3.2.2; Carton et al. 2018). The observational datasets used for this study span from 1980 to 2018, except for ERSST, which covers the period from 1900 to 2018.

In this study, we employ regression and composite analyses as the primary tools to investigate ENSO features. We represent El Niño using indices over crucial regions, including Niño 3 (5°N–5°S, 150°W–90°W), Niño 3.4 (5°N–5°S, 170°W–
120°W), and Niño 4 (5°N–5°S, 160°E–150°W), in the regression analysis. For the observational data, we use a base period between 1971 and 2000, following the Niño index calculation of the Climate Prediction Center, NOAA. In contrast, we use model data from 1900 to 2014 as the base period for TaiESM1's historic run. To avoid impacts of model bias on longer timescales, such as interdecadal variation, we utilize the full length of available simulation data to obtain the most robust statistics of ENSO feature simulated by TaiESM1.

We choose the composite method instead of the regression map to better identify teleconnection signals associated with the El Niño events. In our preliminary analysis, the regressed maps of El Niño events show similar patterns with the composite events in the tropics, but with much weaker signals in the midlatitudes (not shown). Furthermore, as the ENSO events simulated in TaiESM1 shows very symmetric alternations between El Niño and La Niña events, our composite based on El Nino event is followed by La Niña event. To build the El Niño composite, we choose strong events with the Niño 3.4 index larger than 1
standard deviation (i.e., 1.22°C) over the simulation period. The ERSSTv5 dataset includes eight El Niño events in 1982, 1986, 1987, 1991, 1994, 1997, 2002, and 2009. In comparison, TaiESM1 simulated 21 events throughout the entire historical simulation. As the El Niño events simulated by TaiESM1 exhibit very strong amplitude, most of the composite fields of these events passed the significance test at the 95% confidence level (not shown). Therefore, we will not denote regions passing significant tests in the composite fields in the following analysis.

## 115  3 Basic statistics of ENSO in TaiESM1

### 3.1 Niño 3.4 SST variability

Figure 1 shows the mean SST state (white contour) and monthly standard deviation (color shading) in ERSST and TaiESM1 over the tropical Pacific. The monthly standard deviation denotes the deviation of monthly SSTs from long-term monthly mean SST. TaiESM1 has a much more substantial equatorial Pacific SST variability, with an elongated region of high
SST variation extending further into the warm pool region. Such an westward extension is collocated with the equatorial cold tongue, indicated by the 27°C isotherm thick white contour laying over the tropical eastern Pacific in the climatological SST mean field. TaiESM1 simulated a westward extension of the cold tongue compared with ERSSTv5. TaiESM1 also overestimated SST variation over other tropical oceans, including the Indian Ocean and warm pool.



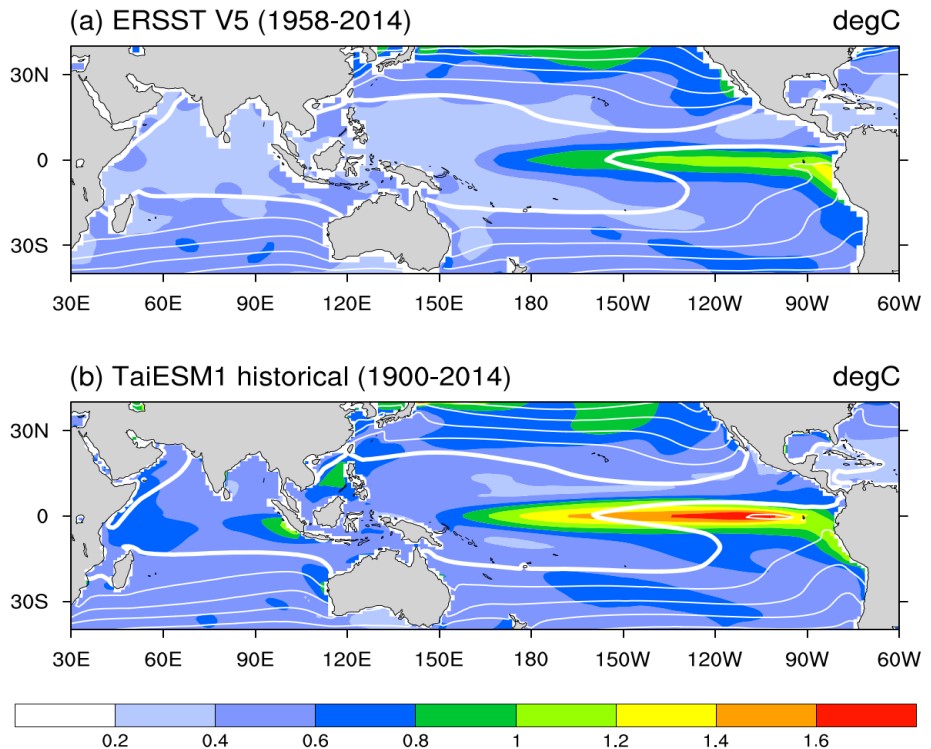

**Figure 1: Mean SST (white contours) and SST monthly standard deviation (color shading) for (a) detrended ERSSTv5 over 1958–2014 and (b) 1900–2014 of the TaiESM1's historical simulations. The contour interval is 3°C, and the thick white contour indicates the 27°C isotherm.**

Figure 2 shows the Niño3.4 SST index based on the ERSSTv5 and TaiESM1 historical simulations. TaiESM1 has more oscillatory ENSO signals with alternating cold and warm phases during 3–5 years compared with the observations. The Niño 3.4 index's standard deviation of ERSSTv5 is 0.84°C, whereas that of TaiESM1 is 1.22°C. The simulated ENSO amplitude decreased after 1980 when the global temperature increased (Fig. 2a, b). The power spectrum of Niño 3.4 confirms what is found in the time series of Niño 3.4, namely a much larger amplitude of ENSO in TaiESM1. The amplitude of major peak

between 3 and 4 years is around $250°C^2$/month, while that of observed peak is around $75°C^2$/month in ERSSTv5. Similarly, around the secondary peak with period of 5 to 6 year in observations, TaiESM1 also shows two spectral peaks at 6 year and 8 year with stronger amplitude, respectively. Such model bias in representing ENSO-related spectral peaks have long noticed in CMIP models and still one of most challenging questions for climate models (Jha et al., 2014). Fig. 2d shows that the seasonal cycle of ENSO SST variance in ERSSTv5 and TaiESM1. Compared with the observations, the peak months simulated by

TaiESM1 occurred in boreal winter, with one month delay than the observed, and a larger amplitude that was 1.5 times of the observed value.



**Figure 2: Normalized time series of Niño3.4 index in (a) ERSST V5 (1900–2018) and (b) TaiESM1 historical run (1900–2014). The standard deviation of each dataset is noted on the upper-right side of the panels. The corresponding spectrums are shown in (c), with a black line for ERSSTv5 and a blue line for TaiESM1. (d) The seasonal cycle of SST variance in ERSSTv5 (black) and TaiESM1 (blue).**





### 3.2 Atmosphere–ocean coupling of ENSO

As a coupled oscillation system, the coupling of atmosphere and ocean plays an important role in ENSO dynamics. To
see how such coupling is simulated in TaiESM1, Figure 3 shows the regressed rainfall, wind stress, and SST to the Niño 3.4
index in the observations (Fig. 3a) and TaiESM1 (Fig. 3b). In Fig. 3a, the observation shows the west–east displacement of
wind stress and warm SST over the tropical Pacific. The strong wind stress at 160°E is collocated with rainfall due to (mostly
meridional) moisture convergence at the west and north edges of the warm SSTA, which was located in the central-eastern
equatorial Pacific and was approximately 1° higher than the western equatorial Pacific. It is important to note that the major
sea surface temperature anomaly (SSTA) did not occur in the eastern equatorial Pacific, where interannual variance was the
highest. Instead, the SSTA occurred to the west of the region with the maximum variance. In TaiESM1 (as shown in Fig. 3b),
the warm SST center is located approximately at 120°W, which is further east than in the observational data. Moreover, the
magnitude of the SST anomaly is approximately 50% greater in TaiESM1 than in the observations. Compared with ERSST,
the warm SSTA in TaiESM1 is meridionally narrower and more zonally elongated to the western Pacific around 155°E,
causing stronger zonal and meridional SST gradients. Increase of the meridional SST gradient induces stronger meridional
wind and moisture convergence over the equatorial Pacific. Zonally, the stronger westerly wind extends wider from 155°E to
120°W near the eastern edge of New Guinea Island. As a result, TaiESM1 produces strong wind stress and more deep
convection (shown by rainfall; color shading in Fig. 3) to the north and west sides of the warm SSTA.    Figure S1 shows the
regressed magnitude of wind stress onto the Niño3.4 index and marks longitude center with dashed lines. While TaiESM1
reproduces the longitudinal center of wind stress response at 140°E as in observations, the response magnitude of wind stress
to SST increase is weaker in TaiESM1 than in the observations.

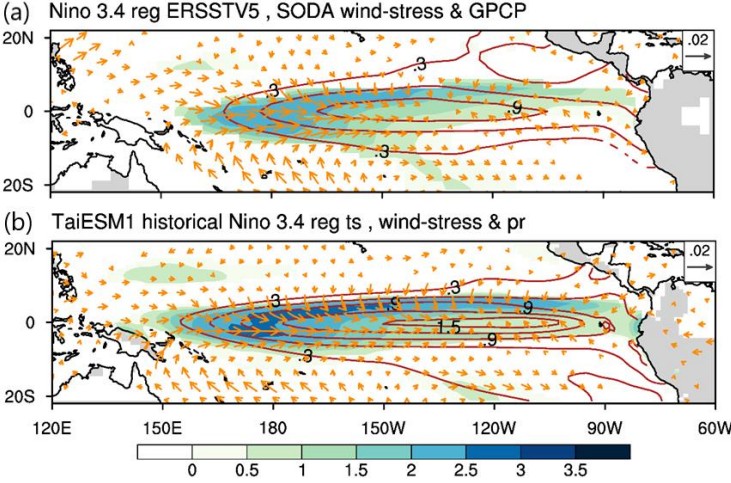

**Figure 3: Regression map of precipitation (mm/day, color shading), wind stress (1/s, vectors), and sea surface temperature (SST; °C, contours) on the normalized Niño-3.4 index for (a) Global Precipitation Climatology Project (GPCP), Simple Ocean Data Assimilation (SODA), Extended Reconstructed SST (ERSST), and (b) TaiESM1 historical simulation.**

Figure 4 shows the regressed SSH to the wind stress averaged over the Niño-4 region (5°N–5°S, 150°E–90°E) to show the thermocline response to the strengthening of equatorial wind stress over the western Pacific. In the observation, a west–
east dipole of thermocline response is found in Fig. 4a, showing thermocline deepening over the east Pacific (marked as black square in Fig.4) and shallowing over the western subtropical Pacific. Compared with the observations, the TaiESM1 has captured this west–east dipole of SSH response to equatorial wind stress, but with a much stronger magnitude over the eastern equatorial Pacific (Fig. 4b). Such a strong response indicates that the SSH in TaiESM1 is more responsive than the observed to the wind stress and can easily lead to an El Niño state through the Bjerknes feedback by reducing the zonal SST gradient
when the wind stress anomaly in the central equatorial Pacific initiates.

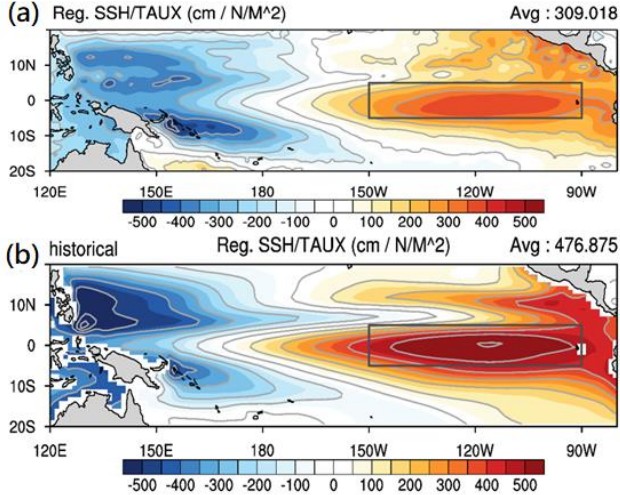

**Figure 4: Regression map of sea surface height (SSH, cm/N/m2, color shading) upon the normalized wind stress averaged over the Niño-4 region of 5°S–5°N and 160°E–160°W (a) from Simple Ocean Data Assimilation (SODA) and (b) the historical run of TaiESM1.**

In addition to the two components of atmosphere-ocean coupling related to the wind stress, we also examine the heat flux-SST coupling related to ENSO. Figures 5a and b show the shortwave radiation regressed to the Niño 3.4 index in MRE2 and TaiESM1. In the observations, the reduction in shortwave fluxes prevails in the tropics with the increased of warm SSTA
of the Niño 3.4 region because of emerging deep convection reflecting more shortwave radiation back (Fig. 5a). A zonal gradient of shortwave fluxes is shown from -160 W m$^{-2}$K$^{-1}$ over the western Pacific (i.e. 170°E) to -80 W m$^{-2}$ K$^{-1}$ over the eastern Pacific (i.e. 120°W). Overall, TaiESM1 reproduced the negative feedback patterns in the deep tropics (3°S−3°N), but with a much stronger shortwave reduction of -200 W m$^{-2}$ K$^{-1}$ over the west Pacific in response to the warm SSTA of the Niño 3.4 region (Fig. 5b). Such pattern is consistent with stronger rainfall response of TaiESM1 in Fig. 3b, suggesting the stronger



deep convection response reflects more shortwave fluxes over the west Pacific. On the other hand, increased downwelling shortwave flux to the Niño 3.4 SST is found near the negative region's two fledges in TaiESM1 and the observations. One notable feature in TaiESM1 is the shortwave fluxes can increase up to 60 Wm$^{-2}$ K$^{-1}$ over the tropical eastern Pacific (i.e. 120°W to 100°W), in contrast to a decrease in observations (Fig.5). Such difference suggests there is a biased cloud radiative response over the eastern Pacific when El Niño event occurs, which may induce biased heat flux-SST coupling in TaiESM1. We will

further examine and discuss this bias in Section 4.

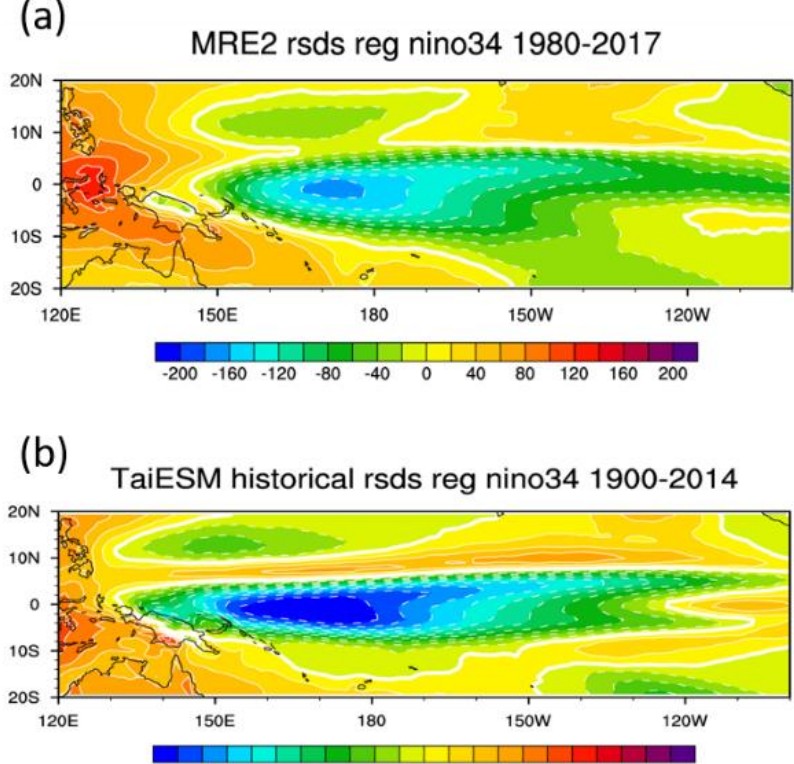

**Figure 5: Regression map of tropical surface downwelling shortwave radiation (RSDS; W/m$^2$; color) upon the normalized Niño-3.4 index for (a) MRE2 ensemble (1980–2017) and (b) TaiESM1 historical simulation (1900–2014).**





### 3.3 Composite of El Niño structure and teleconnection

To evaluate the structure of El Niño events in TaiESM1, we compose the strong El Niño events with the Niño 3.4 index
larger than 1 standard deviation of the entire time series (i.e., larger than 1.22°C). Under this definition, there are 21 Niño
events in TaiESM1 historical run, and nine events in the MRE2 ensemble from 1980 to 2015.

Figures 6 and 7 present the seasonal variation of El Niño events in the tropic and its teleconnection pattern in the mid
latitudes in the MRE2 ensemble and TaiESM1 by showing the 2m surface temperature (color shading; Fig. 6), sea-level
pressure (SLP; contours; Fig. 6), precipitation (color shading; Fig. 7), and 300-hPa stream function (contours; Fig. 7). Overall,
TaiESM1 reproduced the observed spatial structures and teleconnection patterns associated with El Niño; however, consistent
with the over-simulated El Niño signals, TaiESM1 produces much stronger tropical SST warming and teleconnection in
extratropical regions than the observations in all four seasons. As early as in June, July, and August in the first year (JJA$^0$),
TaiESM1 already simulates an SST anomaly with 2°C over the eastern Pacific (Fig. 6a, e) with clear rainfall response in central
Pacific (Fig. 7a, e). A zonal dipole of surface temperature and rainfall between the eastern and western Pacific forms earlier
than in the observations (color shading in Fig. 6a, e and Fig. 7a, e). In SON$^0$, the warm SST anomaly grows even stronger and
expands over the entire tropical Pacific in TaiESM1. As a result, very clear teleconnection similar to that of DJF$^{+1}$ can already
be found in the north hemisphere, including horseshoe-shaped cooling in the western Pacific, and those over the Eurasia and
United States (Fig.6b, f). In the meantime, TaiESM1 captures the responses in the south hemisphere in DJF$^{+1}$, including the
warm and dry response over northern South America, the opposite responses over the southern South America, and the north–
south dipole between Eurasia and South Asia (Fig. 6c, g). As for the rainfall response, TaiESM1 realistically simulates the
shift of deep convection from the western Pacific to the central Pacific as the warm SST occurs in DJF$^{+1}$; however, the west
shift of tropical SSTA causes the surface temperature response pattern also shift westward, resulting in stronger cooling in the
East and Southeast Asia, even into the Indian ocean. In contrast to the weaker SSTA and surface temperature impacts in the
observations (Fig. 6d) in MAM$^{+1}$, a strong teleconnection pattern in the surface temperature over the extratropical regions
sustained into MAM$^{+1}$ in TaiESM1 (Fig. 6h). Furthermore, the overresponse of the rain band over the Indian Ocean could be
due to the mean rainfall biases simulated in TaiESM1, as noticed by Wang et al. (2021). In terms of the atmospheric circulation
anomaly, TaiESM1 successfully captures the southward Rossby wave propagation from the central Pacific to the southeastern
Pacific (contours in Fig. 6a, e and Fig. 7a, e) in JJA$^0$. From SON$^0$ and into DJF$^{+1}$, teleconnection in TaiESM1 intensifies as the
warm SST over the equatorial Pacific develops into the mature stage of El Niño (Fig. 6b, f; Fig. 7b, f). TaiESM1 reproduces
the Rossby wave train response emitted from the equatorial Pacific into North America and the resulting dipole of surface
temperature over North America during DJF$^{+1}$ (Fig. 6c, g). In line with the stronger temperature and rainfall responses observed
during MAM+1, TaiESM1 exhibits El Nino-related circulation anomalies across the tropics (as shown in Figs. 6d, h and Figs.
7d, h).



**Figure 6: The surface temperature (color shading) and SLP (contours; contour interval is 1 hPa; contours smaller than zero are dashed) of the El Niño composite in (a, e) JJA⁰, (b, f) SON⁰, (c, g) DJF⁺¹, and (d, h) MAM⁺¹ based on the MRE2 ensemble (left column) and TaiESM1 historical (right column).**



**Figure 7: As in Fig. 6 but showing precipitation (mm/day; color shading) and 300 hPa stream function (contours; contour interval of $2 \times 10^6$ m$^2$/s anomalies; contour value smaller than zero is dashed).**






Fig.8 shows the seasonal evolution of ocean subsurface potential temperature averaged over the selected El Niño events. The composites show four seasonal means from $JJA^0$ when an El Niño event was identified to $MAM^{+1}$ in the following year. The green line shows the location of the 20°C subsurface isotherm (Z20) during El Niño events, and the gray line shows the

climatological Z20. During $JJA^0$ in the observations, Z20 deepens in the eastern Pacific and shallows in the western Pacific (Fig. 8a). While the TaiESM1 realistically simulated the climatological Z20 depth, it overestimated the response of subsurface temperatures and simulated a flatter Z20 profile in $JJA^0$ (Fig. 8e). Such an overestimation bias in TaiESM1 continues from the beginning of the ENSO evolution through $SON^0$ and $DJF^{+1}$ (Fig. 8b, c, f, and g). Accompanied by the warm bias over the eastern Pacific, the cold bias developed in $SON^0$ when the cool water started to form in the tropical western Pacific (Fig. 8b–

d, f–h). Such a zonal dipole of subsurface temperature bias in TaiESM1 manifests a basin-wide response of ocean circulations during El Niño events. Moreover, both warming and cooling from the surface to 100 m depth was much more pronounced in the model than in the observation, especially over the eastern Pacific. An unrealistic warming in the central-eastern equatorial Pacific is also notable, reflecting the unrealistic westward extension of positive SSTA. This bias led to an anomalously strong SST gradient up to 2°C between 180°W and 150°W, consistent with the wide-spread strong westerly anomalies. More analysis

is needed to determine which model components are more responsible for these biases.



**Figure 8: Equatorial cross-section (5°S–5°N) of the El Niño composite of the potential temperature anomaly (color shading) in (a, e) JJA[0], (b, f) SON[0], (c, g) DJF[+1], and (d, h) MAM[+1] based on SODA3.3.2 (left column) and TaiESM1 historical run (right column). The gray line shows the climatological 20°C isotherm (Z20), and the green dashed line shows the Z20 at the Niño state.**





## 4 Linking the warm SST bias during El Niño events with simulated seasonal mean states

In this section, we analyze the seasonal life cycle of the strong ENSO signals simulated in TaiESM1 to understand the intense tropical warming anomaly of El Niño events, and link this bias with seasonal mean states. Based on the El Niño events defined in the previous session, we construct the seasonal cycle of El Niño events by plotting the Hovmöller diagram averaging over the equator between 3°S to 3°N.

Fig. 9 shows the Hovmöller diagram of SST anomalies and 1000 hPa zonal winds along the equator (3°S–3°N) based on the strong El Niño event selected from the ERSSTv5 and TaiESM1 historical run. In the observation in May, the warm SSTA occurs over the dateline and the westerly wind anomaly starts to propagate to 135°W. In $JJA^0$, the warm SST slowly develops at 180°–135°W, progressively propagates to the eastern equatorial Pacific with westerly anomalies, and reaches maximum amplitude in November, December, and January (Fig. 9a). In contrast, the warm SST when initiated in May intensifies almost simultaneously in the basin east of 150°W in TaiESM1. The warm anomaly quickly reaches 2°C over the entire eastern equatorial Pacific through $JJA^0$ and into September. Such warming seems to be coupled with one branch quickly extending to the eastern equatorial Pacific after the initiation in May whereas the westerly anomaly's major branch is well coupled with SSTA over west Pacific most of time. The warming continues developing even when the westerly anomalies in the eastern equatorial Pacific weakened in $JJA^0$ and reaches maximum in January as observations. This early development of a warm SSTA in the eastern equatorial Pacific in May and continuous warming in $JJA^0$ and $SON^0$ are two primary model bias that may contribute to strong El Niño events in TaiESM1.

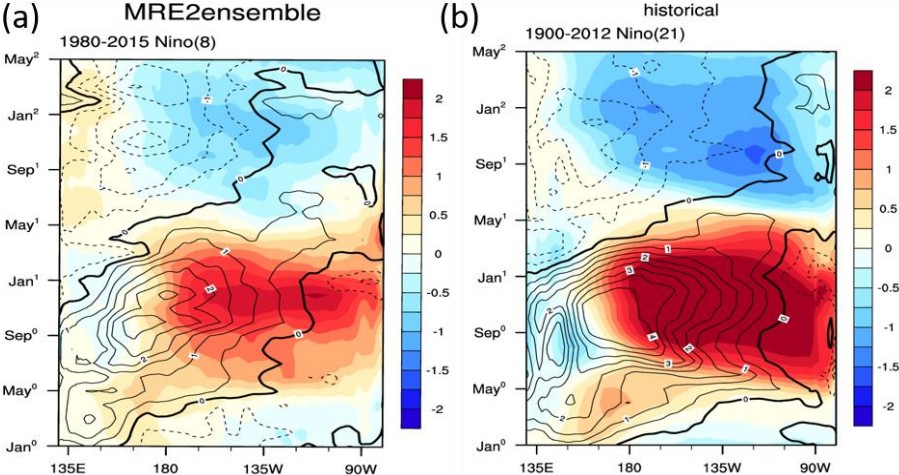

**Figure 9: The Hovmöller diagram of composite SST anomalies (color) and zonal wind on 1000 hPa (contour) along the equator (3°S–3°N) for El Niño based on (a) observations (ERSST/MRE2 ensemble from 1980 to 2015) and (b) TaiESM1 historical run (1900–2014).**

To understand the heat flux-SST coupling, we examined the similar composite of net surface heat flux (Fig. S2a, b) and latent heat fluxes (Fig. S2c, d), which are the two heat fluxes are prominent in observational El Niño events. Notably, in the observations, the seasonal variation of surface net heat fluxes in MRE2 is primarily controlled by latent heat fluxes, especially





from May to November, as other heat fluxes play a minor role (Fig. S2c, d). However, the spatial patterns of net surface fluxes of TaiESM1 are primarily dominated by the shortwave surface fluxes (Fig. S2d) and amplified by the latent heat fluxes (Fig. 2b). Figure 10 shows the same Hovmöller diagram but for downwelling surface shortwave flux (color shading) and surface temperature (contours) composited over the El Niño events. Evolutions of shortwave radiation and SST during the lifecycle of El Niño events in observation and TaiESM1 are shown in Figs. 10a and b, respectively. In the observation, about 10 to 20 W/m$^2$ shortwave fluxes going upward (i.e. shown as negative on Fig.10) during the entire El Niño periods, which is well collocated with the warm SSTA, due to the increased shortwave reflection of deep convection triggered by warmer SST. Such reflection of shortwave fluxes by deep clouds induces negative feedback (i.e., higher SST, large reflected shortwave radiation) between shortwave radiation flux and SST in the tropics (Fig. 10a). The negative feedback intensified in September, reached the maximum in December, and continued to next February (Fig. 10a). In contrast to the observations, TaiESM1 produced unrealistically strong negative feedback over the western equatorial Pacific and Intertropical Convergence Zone (ITCZ) regions (Fig. 10b), and very strong positive shortwave radiation anomalies near the equatorial eastern Pacific after May in year 0 (Fig. 10a). The increase in shortwave influx starts from the eastern Pacific in May and gradually extends to 125°W in November, a feature that was not seen in observation. The westward progression of shortwave radiation increase contributes to erroneous strong SST warming, and its westward extension is seen in TaiESM1 from May to Jan in year 1 (Fig. 10b), instead of the observed simultaneous warming in the eastern Pacific.

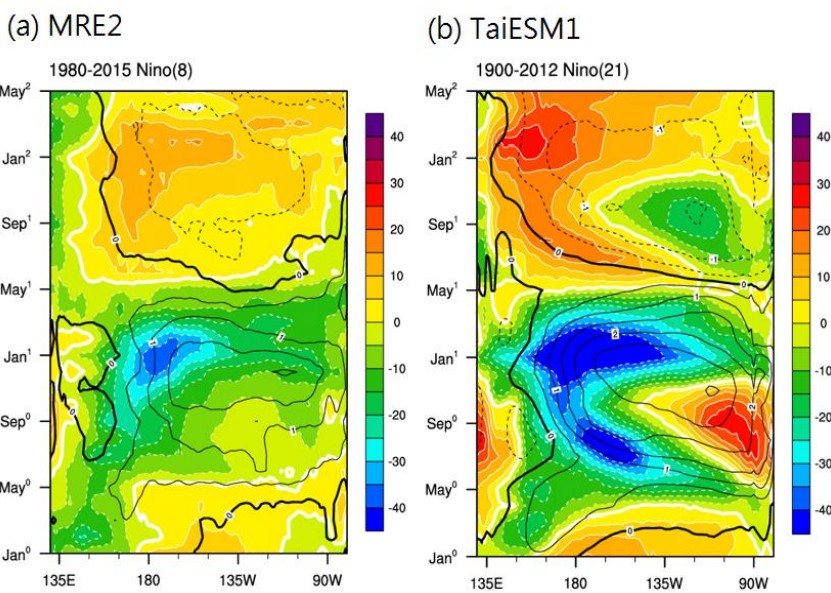

**Figure 10: A composite of El Niño events of RSDS (W/m$^2$; color) and surface temperature (°C; contour) in (a) MRE2 and (b) TaiESM1. Both variables are averaged over the equator (3°S–3°N).**

To understand the cause of the strong SSTA development from May to November identified in Fig. 9 and its relationship with shortwave radiation fluxes in Fig.10, we analyzed the seasonal cycle of low clouds (gray contours), vertical velocity





(green contours), and SST (color shading) in the observations and TaiESM1 in Fig. 11a and b. Figure 11c shows the differences between TaiESM1 and MRE2. Figure 11d shows the climatological seasonal SST cycle at 100°W to show the SST differences where cloud biases are prominent. On the seasonal timescale in observations, the SSTA is closely coupled with anomalies of

vertical velocity and low-level clouds. Cold surface temperature is commonly collocated with excessive low-level clouds and subsidence (Fig. 11a). While TaiESM1 also exhibits a clear seasonal variation as the observations in the east tropical Pacific (Fig. 11b), the simulated seasonal cycle is rather asymmetric with colder bias during $MAM^0$ and $SON^0$ and warm biases during $JJA^0$ and $DJF^{+1}$ (Fig. 11c). Compared with the observations, TaiESM1 warmed approximately one month later during $MAM^0$ and cooled deeper over the eastern Pacific in $SON^0$ (Fig. 11c). The warmer $MAM^0$ sea surface in TaiESM1 provides a smaller

zonal SST gradient and may lead to the earlier onset of the east propagating westerly anomaly found in $MAM^0$ through the Bjerknes feedback (Fig. 9b). However, this cold surface temperature bias during $SON^0$ provides a cold lower boundary for low stratus clouds to develop, leading to an environment for the strong positive feedback between shortwave fluxes and SST during El Niño events. Such impacts of cold tongue bias are also found in CESM1 and CESM2, which shares the same ocean model (i.e., POP2) as TaiESM1 (Wei et al., 2021; Wang et al., 2015a).

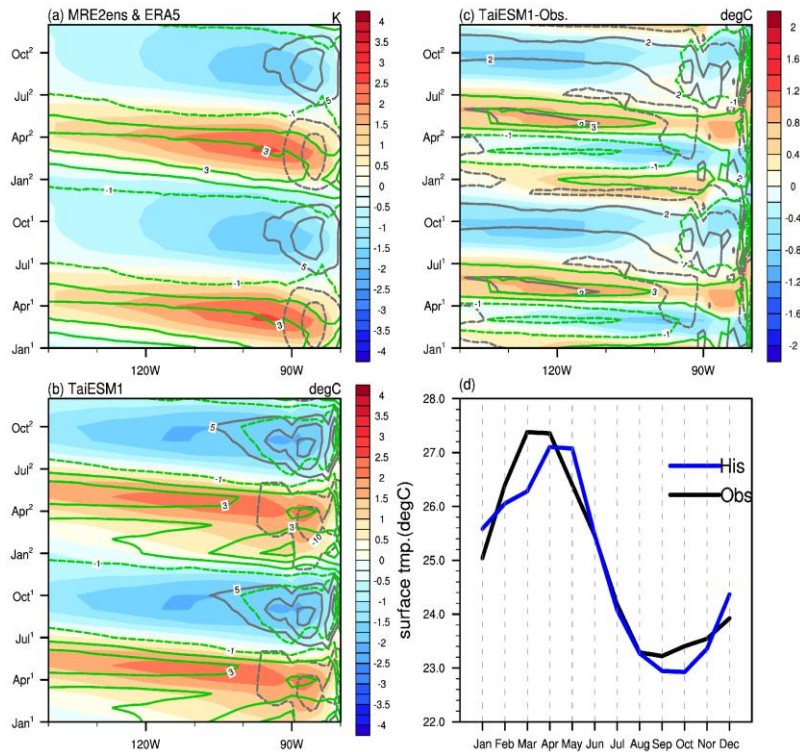

**Figure 11: Climatological seasonal cycle of the surface temperature (shaded area; °C), low clouds (gray contours; %), and 500 hPa vertical velocity (green contours; hPa/s) of the tropical Pacific (3°S–3°N) in the (a) MRE2 ensemble and (b) TaiESM1, and (c) their differences. (d) The seasonal cycle of surface temperature at 100°W in MRE2 (black) and TaiESM1 (blue).**




To understand further look into the bias in SON$^0$, Fig. 12 shows the changes in cloud fraction and zonal circulations in SON$^0$ during the composited El Niño events. TaiESM1 successfully reproduced the eastward shift of the upward branch related to deep convection in response to the warm SST during El Niño events. However, TaiESM1 produced a much stronger response in circulations and cloud cover than the observations (Fig. 12b). Especially over the eastern Pacific, stronger upward motion

anomaly occurs and about 10%–20% of low clouds is reduced in TaiESM1, indicating that a dramatic reduction of low-cloud regime (Fig. 11c). As a result, more shortwave heat influxes are allowed into the ocean surface in SON$^0$ when El Niño events occur (Fig. 10b). Consistent with the seasonal variation of SSTA in Fig.9b, the increased shortwave fluxes thus help to keep the warm SSTA over the SON$^0$ in the east Pacific after the warm SSTA initiates in MAM$^0$ in TaiESM1 (Fig.10b).

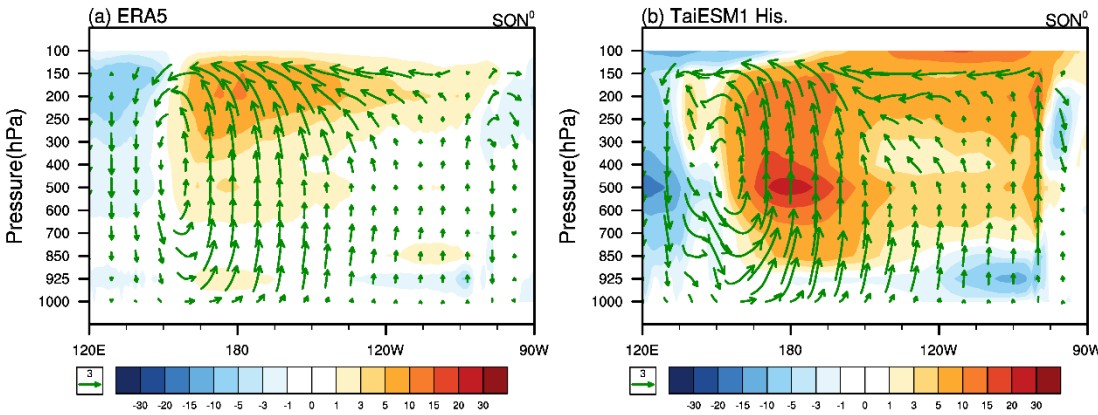

**Figure 12: Longitudinal height cross-section of the El Niño composite of September, October, and November in year 0 (SON$^0$) along the equator (5°S–5°N) based on (a) ERA5 and (b) TaiESM1. The cloud fraction (%) of the El Niño composite is shown in color shading, and the zonal and vertical winds (units: hPa/s) are shown as green vectors.**

## 5 Summary

This study documented ENSO's fundamental statistics and features in the TaiESM1, a CMIP6 participant. Compared with observational dataset, TaiESM1 has captured many prominent observed ENSO features, including a 3–5-year spectrum peak, seasonal phasing, the evolution of warm SST, deepening of the subsurface layer during the El Niño event, and teleconnection patterns in midlatitudes. However, the simulated El Niño signals in TaiESM1 are much stronger and more prolonged than the observed ENSO signals. Such strong signals are shown as intense warm SSTA over the tropical Pacific within the spatial

structure of the composite of the El Niño event. In the meantime, in response to the tropical warm SSTA, the teleconnection wave activities associated with El Niño events are much stronger in TaiESM1, significantly affecting temperature and rainfall in high latitudinal regions, although with a similar spatial pattern to the observations.

To understand the cause of the warm El Niño biases, we investigated the seasonal cycle of strong El Niño events in TaiESM1. Compared with the observed delay propagation of SST after the wind anomaly's initiation to the central Pacific in

May, the simulated SST warming quickly propagates toward the eastern Pacific along with the westerly wind anomaly in





TaiESM1. The SST warming continued through to November and reached a maximum in December, showing a stronger and more prolonged warm period than the observation. Moreover, in contrast to the negative feedback found in observations, our analysis shows that the strong El Niño warm anomalies in TaiESM1 is due to the strong SST-heat flux positive feedbacks, especially over the eastern Pacific in SON[0]. Further analysis shows that this biased feedback is due to the response of spurious
low-cloud regime outside the west coast of South America simulated in TaiESM1. This biased cloud regime results from the seasonal variation of the cold tongue over the east Pacific, consistent with previous studies using the CESM family and CMIP models  (Ham and Kug, 2012; Wei et al., 2021). During El Niño events, the stratus clouds over the eastern Pacific gradually diminished due to the warmer SST, allowing solar radiation to warm the ocean surface. This result leads to a positive feedback of downward solar radiation and SST over the eastern Pacific, an opposite sign of the observed relationship. This biased
relationship is typical in the CMIP5 and CMIP6 models (Bayr et al. 2019; Beobide-Arsuaga et al. 2021).

In summary, TaiESM1 can reproduce many fundamental features of ENSO. However, it still possesses several biases shared by other CMIP6 models, including the lack of randomness of El Niño events, too strong El Niño magnitudes, and early SST warming in the early stage of El Niño events. The strong El Niño strength, our analysis found, is  mainly resulted from the biased atmosphere-SST coupling accompanied by biases of the mean state and seasonal cycle of the cold tongue and
Walker circulation, which is consistent with previous studies with CMIP5/CMIP6 models (Bayr et al., 2018; Chen et al., 2020). However, to resolve this bias (and others), more detailed analysis, including process-related metrics, and more model experiments, such as atmosphere-only and ocean-only experiments, is required to dissect the cause and effects of the seen biases. Also echoed with other studies of ENSO evaluations of CMIP6, our analysis suggests that a more process-based development strategy focusing on atmosphere–ocean coupling rather than a feature-based evaluation of ENSO is needed to
reduce the uncertainty of ENSO simulations and future ENSO projection in climate models.

**Code and Data availability:**

All observational and analysis datasets used in this study are available online. The El Niño index can be downloaded from the NOAA Climate Prediction Center website (https://www.cpc.ncep.noaa.gov/products/analysis_monitoring/ensostuff/ensoyears_1971-2000_climo.shtml). The MRE2
ensemble can be downloaded from the website of the Collaborative REAnalysis Technical Environment – Intercomparison Project (https://esgf-node.llnl.gov/projects/create-ip/). The ERA5 monthly data can be downloaded from the Copernicus Climate Change Service Climate Data Store (doi: 10.24381/cds.f17050d7; doi: 10.24381/cds.6860a573). The Simple Ocean Data Assimilation (SODA) data v.2.3.3 can be downloaded from the SODA website (http://www.soda.umd.edu/). The Global Precipitation Climatology Project version 2.3 and Extended Reconstructed Sea Surface Temperature version 5 can be
downloaded from the website of the NOAA PSL, Boulder, Colorado, USA (https://psl.noaa.gov/data/gridded/data.gpcp.html and https://psl.noaa.gov/data/gridded/data.noaa.ersst.v5.html). The model code of TaiESM version 1 is available at https://doi.org/10.5281/zenodo.3626654. All post-process codes for producing figures presented in this paper are available at





**Author contribution**

**Yi-Chi Wang, Wan-Ling Tseng:** Methodology, Investigation, Writing - Original draft, Writing – Review&Editing. **Yu-Luen Chen:** Software, Formal analysis, Visualization. **Shi-Yu Lee:** Conceptualization, Investigation, Writing – Review&Editing **Huang-Hsiung Hsu**: Conceptualization, Methodology, Writing – Review&Editing, Supervision. **Hsin-Chien Liang:** Data curation.

**Competing interests**

The authors have no relevant financial or non-financial interests to disclose.

**Acknowledgements**

This work was supported by the Taiwan Ministry of Science and Technology under grant numbers MOST107-2111-M-001-010, MOST 109-2111-M-001-012-MY3, and MOST 109-2123-M-001-004.

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
