# Peer review of "ENSO statistics, teleconnections, and atmosphere-ocean coupling in the Taiwan Earth System Model version 1"

_Geoscientific Model Development, 2023_

## Author Comment (AC1)

Dear Reviewer,

We would like to express our sincere gratitude for your valuable comments and suggestions on our manuscript titled **" ENSO statistics, teleconnections, and atmosphere-ocean coupling in the Taiwan Earth System Model version 1"**. In response to your comments, we have carefully addressed each point and made two major revisions:

1.  (Question#3) In response to the comment, we have conducted an analysis on ENSO diversity in TaiESM1 by dividing the simulated El Niño events into Central Pacific El Niño (CP) and East Pacific El Niño (EP) based on the Niño3-Niño4 indices (Kug et al., 2009). Our analysis revealed that TaiESM1 has overestimated the frequency of CP El Niño events compared to observations. However, it captures a similar teleconnection spatial pattern when compared to observations (see Figure R1). These findings have been added to the revised manuscript, providing a more comprehensive understanding of the ENSO diversity in TaiESM1.
2.  (Question#4) To investigate the impacts of the background state on ENSO variability in TaiESM1, we conducted an analysis comparing the changes in the background state between two 30-year time periods, namely 1984-2014 and 1950-1980. Our findings indicate that TaiESM1 exhibits a La Niña-like background during the 1984-2014 period, when weaker ENSO variability prevails, consistent with previous literature (see Figure R2). These results have been included in the revised manuscript to enhance our understanding of the relationship between the background state and ENSO variability in TaiESM1.

Please view the attached pdf file for the complete response report. Thanks again for your comments in improving the quality and clarity of this research.

Sincerely,
Yi-Chi Wang and Coauthors

RC1: 'Comment on gmd-2023-41', Anonymous Referee #1, 10 Apr 2023  reply
        This paper evaluates the Taiwan Earth System Model version 1 (TaiESM1), a recent addition to the class of CMIP models, against various data sets.  The model is shown to have a too strong and regular ENSO cycle similar to the model is it derived from (CESM). The model also exhibits the usual systematic errors, like a cold tongue bias and a positive SST-SW radiation feedback in the eastern Pacific which the authors argue accounts for many of the biases in the model ENSO cycle.  The paper will be useful addition to the literature for those interested in the analysis of CMIP models, particularly their ENSO variability.  There are a few issues that that authors should address in a revision of the manuscript, listed in order of appearance in the paper. The most significant issues are raised in points #3 and #4.

1.  Lines 82-83.   Is this the resolution of the ocean component, the atmospheric component, or both?

Thank you for the question. In TaiESM1, the atmospheric component has a resolution of 0.9° latitude × 1.25° longitude, as stated in the manuscript. In the meantime, the ocean component has a grid resolution of approximately 1.125° in longitude and 0.47° in latitude.

We have revised the text as:

"The historical run is conducted based on the pre-industrial control run of TaiESM1. It utilizes an atmospheric model with a horizontal resolution of 0.9° latitude × 1.25° longitude and 30 vertical layers. The community land model employed in the historical run shares the same resolution as the atmospheric model. Additionally, the POP2 ocean model has a resolution of approximately 1.125° in longitude and 0.47° in latitude."

2. Line 102. Why did you use a base period for the model that was different than for the data? What are the differences between the model base period used and a 1970-2000 base period?

Thank you for your question. After double-checking our analysis, we found that we misspelled the base period of TaiESM1 as 1970-2000 in the manuscript. The correct base period for TaiESM1 aligns with the ERSSTV5 dataset, spanning from 1900 to 2014.

The revised manuscript now accurately reflects the base period for both TaiESM1 and the ERSSTV5 dataset as:

"For the observational Niño 3.4 index, we use a base period between 1900 and 2014 from ERSSTV5, following the Niño index calculation of the Climate Prediction Center, NOAA. In contrast, we use model data from 1900 to 2014 as the base period for TaiESM1's historic run."

3. The authors use a composite of eight observed ENSO events (lines 110-111) to compare with the model output. However this set is comprised of a combination of eastern Pacific (EP) and central Pacific (CP) El Niños with distinctly different spatial structures (McPhaden et al., 2011; Capotondi et al., 2021). The authors should include a discussion of how well TaiESM1 simulates ENSO diversity as this is one of the most important problems in ENSO research today.

We appreciate your insightful comment. We have analyzed TaiESM1's capability in simulating ENSO diversity, distinguishing between Eastern Pacific (EP) and Central Pacific (CP) El Niño using the Niño3-Niño4 approach (Kug et al., 2009). We acknowledge that 23 CP events and 17 EP events are identified in TaiESM1 historical period, exhibiting a higher frequency of CP events compared to observations, consistent with previous findings in CMIP models (Capotondi et al., 2020; Chen et al., 2017; McPhaden et al., 2011). The model also depicts higher SSTA in both EP and CP scenarios, aligning with the stronger ENSO magnitude noted in our research (Figure R1a).

The composite of surface temperature and SLP was constructed based on four East Pacific (EP) events and four Central Pacific (CP) events during the observation period (1980-2014), as well as CP and EP events during the historical TaiESM1 period (1900-2014). In terms of the EP composite, TaiESM1 successfully captures the observational features over the tropical Pacific, as demonstrated in Figs. R1b and R1c. However, the CP events identified in TaiESM1 exhibit elongated warm sea surface temperature anomalies in the tropical region, but with weaker teleconnections to the midlatitude in the northern hemisphere (Figs. R1d and R1e). Additionally, the warming over North America is less pronounced and retreats towards the polar region, whereas

the observed cold surface temperature anomaly is replaced by a warm anomaly. It is likely that the biases in the model's mean state contribute to the model's biases in ENSO diversity and teleconnection patterns observed in TaiESM1, which is a common issue seen in other climate models as well (Ham & Kug, 2012).

The discussions of ENSO diversity are now incorporated into the revised manuscript.

[Figure]

**Figure R1:** (a) Composites of equatorial SSTA profiles averaged in 5°S-5°N for EP (red line) and CP (blue line) events identified in ERSST5 (solid line) and in TaiESM1 simulations (dashed line). (b-c) Composites of surface temperature and SLP of MRE2 ensemble based on EP and CP events. (d-e) Composites of surface temperature and SLP of TaiESM1 historical runs based on EP and CP events.

4. Lines 127-28. The comment about diminishing ENSO amplitude is interesting but not further elaborated on. Is the background state in the model changing like in observations, i.e. becoming more La Nina like? We know changes in background state affect ENSO (Fedorov et al., 2021; Cai et al, 2021). This sentence warrants further elaboration since ENSO in a changing climate is also one of the most important problems in ENSO research today.

Thank you for your suggestion. To understand the impact of the model's background state on the ENSO variability changes in TaiESM1, we have compared sea surface temperature (SST) and surface winds between the two 30-year periods of 1950-1980 and 1984-2014 (Fig. R2). TaiESM1 exhibits stronger ENSO variability during 1950-1980 and weaker ENSO variability during 1984-2014 (Fig.2).

Fig. R2 reveals a shift in the background state to a La Niña-like state with increasing zonal temperature gradient over the tropical Pacific and strengthening of the trade winds during 1984-2014, compared to the period of 1950-1980. Previous researches on the ENSO response to changes in the observed mean state indicates that such an increase in zonal wind stress could cause a weakening of feedbacks related to El Niño (Fedorov et al., 2020; Zhao & Fedorov, 2020). This aligns with the observed weakening of ENSO variability in TaiESM1 during 1984-2014.

We have incorporated these discussions on the changes in the background state into the revised manuscript and added figure R2 to our supplementary figures.

[Figure]

**Figure R2:** Difference of sea surface temperature (color shading) and 1000-hPa winds (arrows) during December, January, and February (DJF) between the two periods of weak ENSO variability (i.e. 1984-2014) and strong ENSO variability (1950-1980).

Line 176.  I don't understand the meaning of "fledges" as used here.

Apologies for the confusion caused by the term "fledges." In this context, we were referring to the regions of subsidence adjacent to the ITCZ region (Fig. R3; marked in red squares). Therefore, the revised sentence would read: "In contrast, we observed an increase in downwelling shortwave flux over the subsidence regions adjacent to the ITCZ in both 10ºN and 10ºS over the east Pacific."

[Figure]

**Figure R3:** The regression map of downwelling shortwave flux onto the Niño 3.4 in (a) MRE2 and (b) TaiESM1. Red squares show where SST-shortwave positive feedback occurs.

5.  Lines 321-25.  The authors describe what needs to be done to resolve the causes of the biases in this model.  But they don't say that the needed actions will actually be taken. Is there a plan to carry out more analyses to resolve the problems?

Thank you for raising this important point. Indeed, further actions are planned to address the identified biases in our model. We have added this information to our manuscript to provide a clearer outlook on our future work. Our forthcoming research will focus particularly on the two model biases related to ENSO in TaiESM1. To do this, we plan to implement ocean-only experiments with the ocean component POP2, allowing us to quantify the ocean's response to biased winds and radiation fluxes. At the same time, we will conduct AMIP-type simulations to investigate the development of westerly wind anomalies under biased SST conditions. This exploration will give us valuable insights into the influence of fast-propagating westerly wind anomalies on the formation of El Niño events. Combined with process-oriented diagnosis for these

model experiments, this approach will allow us to dissect and better comprehend the causes and effects of these observed biases.

The discussion section of our manuscript has been revised to reflect these points.

**References**

Cai, W., Santoso, A., Collins, M., Dewitte, B., Karamperidou, C., Kug, J.-S., Lengaigne, M., McPhaden, M. J., Stuecker, M. F., Taschetto, A. S., Timmermann, A., Wu, L., Yeh, S.-W., Wang, G., Ng, B., Jia, F., Yang, Y., Ying, J., Zheng, X.-T., … Zhong, W. (2021). Changing El Niño–Southern Oscillation in a warming climate. *Nature Reviews Earth & Environment*, *2*(9), 628–644. https://doi.org/10.1038/s43017-021-00199-z

Capotondi, A., Wittenberg, A. T., Kug, J., Takahashi, K., & McPhaden, M. J. (2020). *ENSO Diversity* (pp. 65–86). https://doi.org/10.1002/9781119548164.ch4

Chen, C., Cane, M. A., Wittenberg, A. T., & Chen, D. (2017). ENSO in the CMIP5 Simulations: Life Cycles, Diversity, and Responses to Climate Change. *Journal of Climate*, *30*(2), 775–801. https://doi.org/10.1175/JCLI-D-15-0901.1

Fedorov, A. V., Hu, S., Wittenberg, A. T., Levine, A. F. Z., & Deser, C. (2020). *ENSO Low-Frequency Modulation and Mean State Interactions* (pp. 173–198). https://doi.org/10.1002/9781119548164.ch8

Ham, Y.-G., & Kug, J.-S. (2012). How well do current climate models simulate two types of El Niño? *Climate Dynamics*, *39*(1–2), 383–398. https://doi.org/10.1007/s00382-011-1157-3

McPhaden, M. J., Lee, T., & McClurg, D. (2011). El Niño and its relationship to changing background conditions in the tropical Pacific Ocean. *Geophysical Research Letters*, *38*(15). https://doi.org/10.1029/2011GL048275

Zhao, B., & Fedorov, A. (2020). The Effects of Background Zonal and Meridional Winds on ENSO in a Coupled GCM. *Journal of Climate*, *33*(6), 2075–2091. https://doi.org/10.1175/JCLI-D-18-0822.1

---

## Author Comment (AC2)

Dear Reviewer,

We would like to express our sincere gratitude for your valuable comments and suggestions on our manuscript titled **" ENSO statistics, teleconnections, and atmosphere-ocean coupling in the Taiwan Earth System Model version 1"**. In response to your comments, we have carefully addressed each point and made two revisions accordingly:

1. (Major) After careful examination, we have acknowledged that there was confusion regarding the direction of total heat flux in El Niño evolution in the previous Figure S2, which has opposite direction with Figure 10 in the main text. To eliminate the confusion, Figure R4 below now displays the total surface fluxes with the direction sign aligned with Figure 10. This revised figure clearly illustrates that the surface heat flux has a strong effect in warming the sea surface. To facilitate a better understanding, it is included as the supplementary figure S5.
2. (Minor#4) In response to the comment, we have enhanced the completeness of our analysis by including information on zonal current in the Figure 8, which provides a more comprehensive picture of the subsurface ocean during El Niño events. The corresponding figure, now referred to as Figure R5, has been added as supplementary Figure S3 in the revised manuscript.

Please view the attached pdf file for the complete response report. Thank you again for your comments in improving the quality and clarity of this research.

Sincerely,
Yi-Chi Wang and Coauthors

**RC2**: 'Comment on gmd-2023-41', Anonymous Referee #2, 05 May 2023

I don't know how this journal exactly works, but this manuscript does not present a new model development, nor the first decription of this model. However, it provides the first detailed analysis of ENSO characteristics of this model. If this falls within the scope of the journal that would be ok then. Overall the quality of the analysis of ENSO characteristics is very good. However, I have one major concern, and a few minor comments.

Major: The TaiESM1 shows a very reasonable ENSO in various measures, but has a too strong magnitude. The authors provide some analysis why this may be the case and point to the solar radiation flux increase in El Niño events due to stratos cloud reduction. While this impact is clearly identified, I think it cannot be argued that this is the mechnism clearly responsible for too strong ENSO amplitude. For example, Fig. S2 clearly shows that the net surface heatfluxes much more strongly oppose the dynamically induced ENSO SST anomalies in the model compare to observations. Therefore, it seems more likely that other positive feedback that lead to stonger westerly wind anomalies in the central Pacific are relevant. More analysis is needed here. Perhaps looking at thermocline structure. Overall, while the shown solar radiation positive feedback is certainly there, other heatflux feedback do overcompensate this, leading to a strong negative net

heatflux feedback. The authors have to make more effort if they want to convince that this solar radiation feedback is working.

Thank you for your suggestions. Upon careful reevaluation of our analysis, we acknowledge that there was confusion regarding the direction of total heat flux in Figure S2, leading to a potential misinterpretation of an opposing mechanism for the positive feedback between shortwave-SST that we proposed in our manuscript. We sincerely apologize for any confusion caused.

To ensure clarity and facilitate better understanding, we have reverted the sign of Figure S2 to align consistently with Figure 10 as shown below, ensuring coherence throughout the manuscript. As a result of this adjustment, it becomes evident that both Figure S2 and Figure 10 illustrate the effect of total heat flux entering the tropical Pacific region, which corresponds to the observed warm sea surface temperature patterns. Additionally, we have noticed that the evolution of total heat flux aligns more closely with the evolution pattern of shortwave heat fluxes in Figure 10, providing further support for our hypothesis that shortwave flux warming plays a significant role in driving the warm sea surface temperatures in TaiESM1.

We appreciate your keen attention to detail and for bringing these points to our attention.

[Figure]

Figure R4: Similar to Fig9c, (a,b) but for net radiation flux (color shading; net = rsus-rsds+rlus-rlds-hfss-hlfs, W m-2) for MRE2 and TaiESM1, respectively. Rsus represents shortwave upwelling fluxes, rlus as longwave upwelling surface fluxes, rlds as longwave downwelling surface fluxes, hfss as surface sensible heat fluxes, and hlfs as surface latent heat fluxes.

.

Minor:

1. Line 128: Perhaps also quote the observed Niño3.4 standard deviation in the Satellite period (1981 to 2022), which would be substantially larger than 0.84 and closer to the TaiESM1.

   Thank you for the suggestions. Upon calculating the standard deviation of the Niño3.4 index using ERSSTV5 data from 1981 to 2022, we determined that the standard deviation is 0.85, which is similar to the value of 0.84 obtained when considering the entire period.

2. Fig. 4: What is the box shown and why?

   Sorry for the confusion. The box marks the Niño-4 region where the average taken to calculate regression maps. We have added the description in the caption of Fig.4.

3. Fig. 6: Also mention the dominance of IOD-like response in modelin DJF compared to obs, where we see already the dominance of an IOBM response.

   Thank you for the suggestion. We have updated the discussion in light of the temperature dipole observed over the Indian Ocean in Figure 6g. Specifically, we have added a mention of the dominance of the IOD-like response in the model during DJF, as opposed to the observed data where the dominance of an IOD response is already apparent.

   "however, the west shift of tropical SSTA causes the surface temperature response pattern also shift westward, resulting in enhanced cooling in the East and Southeast Asia. The cooling further extends into the Indian Ocean, causing an Indian Ocean Dipole-like response as depicted in Fig. 6g."

4. Fig. 8: Please provide an additional (supplementary) figure which shows the equatorial temperature structure in a longiture-depth plot to also see the thermocline structure. Perhaps also zonal currents could be shown there.

   Thank you for your suggestion. We have included an additional supplementary figure, Figure R5 below, which presents a longitude-depth plot depicting the equatorial temperature structure (color shading) and zonal current (blue contours). This new figure provides a clear visualization of the thermocline depth during El Niño events, represented by a green line.

   Upon examining this plot, it becomes evident that in TaiESM1, as the El Niño event develops, the westerly current anomaly (solid blue contour) within the thermocline strengthens. This aligns with the stronger sea surface temperature (SST) and warmer subsurface temperature anomaly over the East Pacific region. In TaiESM1, the westerly current anomaly is significantly stronger and extends further east compared to observations in the SODA analysis. While our study does not explicitly investigate the role of zonal advection in contributing to the warming near the East Pacific, it is an interesting aspect that merits further examination. The revised manuscript now includes Figure R5 as Figure S3, enhancing the understanding of the thermocline dynamics and zonal current behavior during El Niño events in TaiESM1.

[Figure]

Figure R5: Equatorial cross-section (5°S–5°N) of the El Niño composite of the zonal current (blue contour) and potential temperature anomaly (color shading) in (a, e) JJA[0], (b, f) SON[0], (c, g) DJF[+1], and (d, h) MAM[+1] based on SODA3.3.2 (left column) and TaiESM1 historical run (right column). The gray line shows the climatological 20°C isotherm (Z20), and the green dashed line shows the Z20 at the Niño state.

---

## Author Response (AR2)

Dear Editor,

Based on the reviewer's suggestion, we have corrected the region indicated in Figure 4 from Niño 4 to Niño 3, as advised (please refer to Line 176 and the caption of Figure 4). This adjustment has been implemented in the current uploaded manuscript titled "ENSO statistics, teleconnections, and atmosphere-ocean coupling in the Taiwan Earth System Model version 1". We are grateful for your valuable help throughout the review process.

Sincerely,

Yi-Chi Wang and Coauthors